# Paraphrasing Complex Network: Network Compression via Factor Transfer

**Jangho Kim**
Seoul National University
Seoul, Korea
kjh91@snu.ac.kr

**SeongUk Park**
Seoul National University
Seoul, Korea
swpark0703@snu.ac.kr

**Nojun Kwak**
Seoul National University
Seoul, Korea
nojunk@snu.ac.kr

## Abstract

Many researchers have sought ways of model compression to reduce the size of a deep neural network (DNN) with minimal performance degradation in order to use DNNs in embedded systems. Among the model compression methods, a method called *knowledge transfer* is to train a student network with a stronger teacher network. In this paper, we propose a novel knowledge transfer method which uses convolutional operations to paraphrase teacher's knowledge and to translate it for the student. This is done by two convolutional modules, which are called a *paraphraser* and a *translator*. The paraphraser is trained in an unsupervised manner to extract the *teacher factors* which are defined as paraphrased information of the teacher network. The translator located at the student network extracts the *student factors* and helps to translate the teacher factors by mimicking them. We observed that our student network trained with the proposed factor transfer method outperforms the ones trained with conventional knowledge transfer methods.

## 1   Introduction

In recent years, deep neural nets (DNNs) have shown their remarkable capabilities in various parts of computer vision and pattern recognition tasks such as image classification, object detection, localization and segmentation. Although many researchers have studied DNNs for their application in various fields, high-performance DNNs generally require a vast amount of computational power and storage, which makes them difficult to be used in embedded systems that have limited resources. Given the size of the equipment we use, tremendous GPU computations are not generally available in real world applications.

To deal with this problem, many researchers studied DNN structures to make DNNs smaller and more efficient to be applicable for embedded systems. These studies can be roughly classified into four categories: 1) network pruning, 2) network quantization, 3) building efficient small networks, and 4) knowledge transfer. First, network pruning is a way to reduce network complexity by pruning the redundant and non-informative weights in a pretrained model [26, 17, 7]. Second, network quantization compresses a pretrained model by reducing the number of bits used to represent the weight parameters of the pretrained model [20, 27]. Third, Iandola *et al*. [13] and Howard *et al*. [11] proposed efficient small network models which fit into the restricted resources. Finally, knowledge transfer (KT) method is to transfer large model's information to a smaller network [22, 30, 10].

Among the four approaches, in this paper, we focus on the last method, knowledge transfer. Previous studies such as *attention transfer* (AT) [30] and *knowledge distillation* (KD) [10] have achieved meaningful results in the field of knowledge transfer, where their loss function can be collectively summarized as the difference between the attention maps or softened distributions of the teacher and the student networks. These methods directly transfer the teacher network's softened distribution [10] or its attention map [30] to the student network, inducing the student to mimic the teacher.

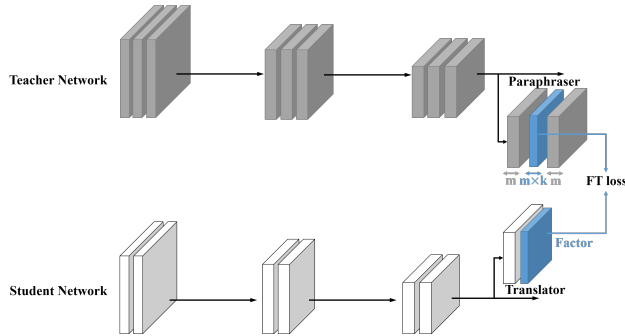

Figure 1: Overview of the factor transfer. In the teacher network, feature maps are transformed to the 'teacher factors' by a paraphraser. The number of feature maps of a teacher network ($m$) are resized to the number of feature maps of teacher factors ($m \times k$) by a paraphrase rate $k$. The feature maps of the student network are also transformed to the 'student factors' with the same dimension as that of the teacher factor using a translator. The factor transfer (FT) loss is used to minimize the difference between the teacher and the student factors in the training of the translator that generates student factors. Factors are drawn in blue. Note that before the FT, the paraphraser is already trained unsupervisedly by a reconstruction loss.

While these methods provide fairly good performance improvements, directly transferring the teacher's outputs overlooks the inherent differences between the teacher network and the student network, such as the network structure, the number of channels, and initial conditions. Therefore, we need to re-interpret the output of the teacher network to resolve these differences. For example, from the perspective of a teacher and a student, we came up with a question that simply providing the teacher's knowledge directly without any explanation can be somewhat insufficient for teaching the student. In other words, when teaching a child, the teacher should not use his/her own term because the child cannot understand it. On the other hand, if the teacher translates his/her terms into simpler ones, the child will much more easily understand.

In this respect, we sought ways for the teacher network to deliver more understandable information to the student network, so that the student comprehends that information more easily. To address this problem, we propose a novel knowledge transferring method that leads both the student and teacher networks to make transportable features, which we call 'factors' in this paper. Contrary to the conventional methods, our method is not simply to compare the output values of the network directly, but to train neural networks that can extract good factors and to match these factors. The neural network that extracts factors from a teacher network is called a *paraphraser*, while the one that extracts factors from a student network is called a *translator*. We trained the paraphraser in an unsupervised way, expecting it to extract knowledges different from what can be obtained with supervised loss term. At the student side, we trained the student network with the translator to assimilate the factors extracted from the paraphraser. The overview of our proposed method is provided in Figure 1. With various experiments, we succeeded in training the student network to perform better than the ones with the same architecture trained by the conventional knowledge transfer methods.

Our contributions can be summarized as follows:

● We propose a usage of a paraphraser as a means of extracting meaningful features (factors) in an unsupervised manner.

● We propose a convolutional translator in the student side that learns the factors of the teacher network.

● We experimentally show that our approach effectively enhances the performance of the student network.

## 2   Related Works

A wide variety of methods have been studied to use conventional networks more efficiently. In network pruning and quantization approaches, Srinivas *et al*. [26] proposed a data-free pruning

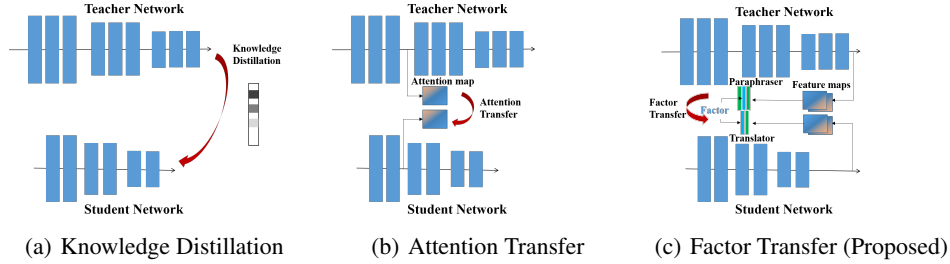

| (a) Knowledge Distillation | (b) Attention Transfer | (c) Factor Transfer (Proposed) |

Figure 2: The structure of (a) KD [10], (b) AT [30] and (c) the proposed method FT. Unlike KD and AT, our method does not directly compare the softened distribution (KD) or the attention map (AT) which is defined as the sum of feature maps of the teacher and the student networks. Instead, we extract factors from both the teacher and the student, whose difference is tried to be minimized.

method to remove redundant neurons. Han *et al*. [7] removed the redundant connection and then used Huffman coding to quantize the weights. Gupta *et al*. [6] reduced float point operation and memory usage by using the 16 bit fixed-point representation. There are also many studies that directly train convolutional neural networks (CNN) using binary weights [3, 4, 20]. However, the network pruning methods require many iterations to converge and the pruning threshold is manually set according to the targeted amount of degradation in accuracy. Furthermore, the accuracies of binary weights are very poor, especially in large CNNs. There are many ways to directly design efficient small networks such as SqueezeNet [13], Mobile-Net [11] and Condense-Net [12], which showed a vast amount of reduction in the number of parameters compared to the original network sacrificing some accuracies. Also, there are methods of designing a network using a reinforcement learning algorithm such as MetaQNN [2] and Neural Architecture Search [33]. Using the reinforcement learning algorithm, the network itself searches for an efficient structure without human assistance. However, they only focused on performance without considering the number of parameters. In addition, it takes a lot of GPU memories and time to learn.

Another method is the 'knowledge transfer'. This is a method of training a student network with a stronger teacher network. Knowledge distillation (KD) [10] is the early work of knowledge transfer for deep neural networks. The main idea of KD is to shift knowledge from a teacher network to a student network by leaning the class distribution via softened softmax. The student network can capture not only the information provided by the true labels, but also the information from the teacher. Yim *et al*. [28] defined the flow of solution procedure (FSP) matrix calculated by Gram matrix of feature maps from two layers in order to transfer knowledge. In FitNet [22], they designed the student network to be thinner and deeper than the teacher network, and provided hints from the teacher network for improving performance of the student network by learning intermediate representations of the teacher network. FitNet attempts to mimic the intermediate activation map directly from the teacher network. However, it can be problematic since there are significant capacity differences between the teacher and the student. Attention transfer (AT) [30], in contrast to FitNet, trains a less deep student network such that it mimics the attention maps of the teacher network which are summations of the activation maps along the channel dimension. Therefore, an attention map for a layer is of its the spatial dimensions. Figure 2 visually shows the difference of KD [10], AT [30] and the proposed method, factor transfer (FT). Unlike other methods, our method does not directly compare the teacher and student networks' softend distribution, or attention maps.

As shown in Figure 1, our paraphraser is similar to the convolutional autoencoder [18] in that it is trained in an unsupervised manner using the reconstruction loss and convolution layers. Hinton *et al*.[9] proved that autoencoders produce compact representations of images that contain enough information for reconstructing the original images. In [16], a stacked autoencoder on the MNIST dataset achieved great results with a greedy layer-wise approach. Many studies show that autoencoder models can learn meaningful, abstract features and thus achieve better classification results in high-dimensional data, such as images [19, 24]. The architecture of our paraphraser is different from convolutional autoencoders in that convolution layers do not downsample the spatial dimension of an input since the paraphraser uses sufficiently downsampled feature maps of a teacher network as the input.

## 3 Proposed Method

It is said that if one fully understands a thing, he/she should be able to explain it by himself/herself. Correspondingly, if the student network can be trained to replicate the extracted information, this implies that the student network is well informed of that knowledge. In this section, we define the output of paraphraser's middle layer, as *'teacher factors'* of the teacher network, and for the student network, we use the translator made up of several convolution layers to generate *'student factors'* which are trained to replicate the *'teacher factors'* as shown in Figure 1. With these modules, our knowledge transfer process consists of the following two main steps: 1) In the first step, the paraphraser is trained by a reconstruction loss. Then, *teacher factors* are extracted from the teacher network by a paraphraser. 2) In the second step, these *teacher factors* are transferred to the *student factors* such that the student network learns from them.

### 3.1 Teacher Factor Extraction with Paraphraser

ResNet architectures [8] have stacked residual blocks and in [30] they call each stack of residual blocks as a 'group'. In this paper, we will also denote each stacked convolutional layers as a 'group'. Yosinski *et al.*[29] verified lower layer features are more general and higher layer features have a greater specificity. Since the teacher network and the student network are focusing on the same task, we extracted factors from the feature maps of the last group as clearly can be seen in Figure 1 because the last layer of a trained network must contain enough information for the task.

In order to extract the factor from the teacher network, we train the paraphraser in an unsupervised way by assigning the reconstruction loss between the input feature maps $x$ and the output feature maps $P(x)$ of the paraphraser. The unsupervised training act on the factor to be more meaningful, extracting different kind of knowledge from what can be obtained with supervised cross-entropy loss function. This approach can also be found in EBGAN [32], which uses an autoencoder as discriminator to give the generator different kind of knowledge from binary output.

The paraphraser uses several convolution layers to produce the teacher factor $F_T$ which is further processed by a number of transposed convolution layers in the training phase. Most of the convolutional autoencoders are designed to downsample the spatial dimension in order to increase the receptive field. On the contrary, the paraphraser maintains the spatial dimension while adjusting the number of factor channels because it uses the feature maps of the last group which has a sufficiently reduced spatial dimension. If the teacher network produces $m$ feature maps, we resize the number of factor channels as $m \times k$. We refer to hyperparameter $k$ as a paraphrase rate.

To extract the teacher factors, an adequately trained paraphraser is needed. The reconstruction loss function used for training the paraphraser is quite simple as

$$\mathcal{L}_{rec} = \|x - P(x)\|^2, \tag{1}$$

where the paraphraser network $P(\cdot)$ takes $x$ as an input. After training the paraphraser, it can extract the task specific features (teacher factors) as can be seen in the supplementary material.

### 3.2 Factor Transfer with Translator

Once the teacher network has extracted the factors which are the paraphrased teacher's knowledge, the student network should be able to absorb and digest them on its own way. In this paper, we name this procedure as 'Factor Transfer'. As depicted in Figure 1, while training the student network, we inserted the translator right after the last group of student convolutional layers.

The translator is trained jointly with the student network so that the student network can learn the paraphrased information from the teacher network. Here, the translator plays a role of a buffer that relieves the student network from the burden of directly learning the output of the teacher network by rephrasing the feature map of the student network.

The student network is trained with the translator using the sum of two loss terms, *i.e.* the classification loss and the factor transfer loss:

$$\mathcal{L}_{student} = \mathcal{L}_{cls} + \beta \mathcal{L}_{FT}, \tag{2}$$

$$\mathcal{L}_{cls} = \mathcal{C}(S(I_x), y), \tag{3}$$

$$\mathcal{L}_{FT} = \|\frac{F_T}{\|F_T\|_2} - \frac{F_S}{\|F_S\|_2}\|_p. \tag{4}$$

With (4), the student's translator is trained to output the student factors that mimic the teacher factors. Here, $F_T$ and $F_S$ denote the teacher and the student factors, respectively. We set the dimension of $F_S$ to be the same as that of $F_T$. We also apply an $l_2$ normalization on the factors as [30]. In this paper, the performances using $l_1$ loss ($p = 1$) is reported, but the performance difference between $l_1$ ($p = 1$) and $l_2$ ($p = 2$) losses is minor (See the supplementary material), so we consistently used $l_1$ loss for all experiments.

In addition to the factor transfer loss (4), the conventional classification loss (3) is also used to train student network as in (2). Here, $\beta$ is a weight parameter and $C(S(I_x), y)$ denotes the cross entropy between ground-truth label $y$ and the softmax output $S(I_x)$ of the student network for an input image $I_x$, a commonly used term for classification tasks.

The translator takes the output features of the student network, and with (2), it sends the gradient back to the student networks, which lets the student network absorb and digest the teacher's knowledge in its own way. Note that unlike the training of the teacher paraphraser, the student network and its translator are trained simultaneously in an end-to-end manner.

## 4 Experiments

In this section, we evaluate the proposed FT method on several datasets. First, we verify the effectiveness of FT through the experiments with CIFAR-10 [14] and CIFAR-100 [15] datasets, both of which are the basic image classification datasets, because many works that tried to solve the knowledge transfer problem used CIFAR in their base experiments [22, 30]. Then, we evaluate our method on ImageNet LSVRC 2015 [23] dataset. Finally, we applied our method to object detection with PASCAL VOC 2007 [5] dataset.

To verify our method, we compare the proposed FT with several knowledge transfer methods such as KD [10] and AT [30]. There are several important hyperparameters that need to be consistent. For KD, we fix the temperature for softened softmax to 4 as in [10], and for $\beta$ of AT, we set it to $10^3$ following [30]. In the whole experiments, AT used multiple group losses. Alike AT, $\beta$ of FT is set to $10^3$ in ImageNet and PASCAL VOC 2007. However, we set it to $5 \times 10^2$ in CIFAR-10 and CIFAR-100 because a large $\beta$ hinders the convergence.

We conduct experiments for different $k$ values from 0.5 to 4. To show the effectiveness of the proposed paraphraser architecture, we also used two convolutional autoencoders as paraphrasers because the autoencoder is well known for extracting good features which contain compressed information for reconstruction. One is an undercomplete convolutional autoencoder (CAE), the other is an overcomplete regularized autoencoder (RAE) which imposes $l_1$ penalty on factors to learn the size of factors needed by itself [1]. Details of these autoencoders and overall implementations of experiments are explained in the supplementary material.

In some experiments, we also tested KD in combination with AT or FT because KD transfers output knowledge while AT and FT delivers knowledge from intermediate blocks and these two different methods can be combined into one (KD+AT or KD+FT).

### 4.1 CIFAR-10

The CIFAR-10 dataset consists of 50K training images and 10K testing images with 10 classes. We conducted several experiments on CIFAR-10 with various network architectures, including ResNet [8], Wide ResNet (WRN) [31] and VGG [25]. Then, we made four conditions to test various situations. First, we used ResNet-20 and ResNet-56 which are used in CIFAR-10 experiments of [8]. This condition is for the case where the teacher and the student networks have same width (number of channels) and different depths (number of blocks). Secondly, we experimented with different types of residual networks using ResNet-20 and WRN-40-1. Thirdly, we intended to see the effect of the absence of shortcut connections that exist in Resblock on knowledge transfer by using VGG13 and WRN-46-4. Lastly, we used WRN-16-1 and WRN-16-2 to test the applicability of knowledge transfer methods for the architectures with the same depth but different widths.

| Student | Teacher | Student | AT | KD | FT | AT+KD | FT+KD | Teacher |
|---|---|---|---|---|---|---|---|---|
| ResNet-20 (0.27M) | ResNet-56 (0.85M) | 7.78 | 7.13 | 7.19 | **6.85** | 6.89 | 7.04 | 6.39 |
| ResNet-20 (0.27M) | WRN-40-1 (0.56M) | 7.78 | 7.34 | 7.09 | **6.85** | 7.00 | 6.95 | 6.84 |
| VGG-13 (9.4M) | WRN-46-4 (10M) | 5.99 | 5.54 | 5.71 | 4.84 | 5.30 | **4.65** | 4.44 |
| WRN-16-1 (0.17M) | WRN-16-2 (0.69M) | 8.62 | 8.10 | 7.64 | 7.64 | **7.52** | 7.59 | 6.27 |

| Student | Teacher | $k=0.5$ | $k=0.75$ | $k=1$ | $k=2$ | $k=4$ | CAE | RAE |
|---|---|---|---|---|---|---|---|---|
| ResNet-20 (0.27M) | ResNet-56 (0.85M) | **6.85** | 6.92 | 6.89 | 6.87 | 7.08 | 7.07 | 7.24 |
| ResNet-20 (0.27M) | WRN-40-1 (0.56M) | 7.16 | 7.05 | 7.04 | **6.85** | 7.05 | 7.26 | 7.33 |
| VGG-13 (9.4M) | WRN-46-4 (10M) | **4.84** | 5.09 | 5.04 | 5.01 | 4.98 | 5.85 | 5.53 |
| WRN-16-1 (0.17M) | WRN-16-2 (0.69M) | **7.64** | 7.83 | 7.74 | 7.87 | 7.95 | 8.48 | 8.00 |

Table 1: Mean classification error (%) on CIFAR-10 dataset (5 runs). All the numbers are the results of our implementation. AT and KD are implemented according to [30].

| Student | Teacher | Student | AT | F-ActT | KD | AT+KD | Teacher | FT ($k=0.5$) | Teacher |
|---|---|---|---|---|---|---|---|---|---|
| WRN-16-1 (0.17M) | WRN-40-1 (0.56M) | 8.77 | 8.25 | 8.62 | 8.39 | 8.01 | 6.58 | 8.12 | 6.55 |
| WRN-16-2 (0.69M) | WRN-40-2 (2.2M) | 6.31 | 5.85 | 6.24 | 6.08 | 5.71 | 5.23 | 5.51 | 5.09 |

Table 2: Median classification error (%) on CIFAR-10 dataset (5 runs). The first 6 columns are from Table 1 of [30], while the last two columns are from our implementation.

In the first experiment, we wanted to show that our algorithm is applicable to various networks. Result of FT and other knowledge transfer algorithms can be found in Table 1. In the table, 'Student' column provides the performance of student network trained from scratch. The 'Teacher' column provides the performance of the pretrained teacher network. The numbers in the parentheses are the sizes of network parameters in Millions. The performances of AT and KD are better than those of 'Student' trained from scratch and the two show better or worse performances than the other depending on the type of network used. For FT, we chose the best performance among the different $k$ values shown in the bottom rows in the table. The proposed FT shows better performances than AT and KD consistently, regardless of the type of network used.

In the cases of hybrid knowledge transfer methods such as AT+KD and FT+KD, we could get interesting result that AT and KD make some sort of synergy, because for all the cases, AT+KD performed better than standalone AT or KD. It sometimes performed even better than FT, but FT model trained together with KD loses its power in some cases.

As stated before in section 3.1, to check if having a paraphraser per group in FT is beneficial, we trained a ResNet-20 as student network with paraphrasers and translators combined in group1, group2 and group3, using the ResNet-56 as teacher network with $k=0.75$. The classification error was 7.01%, which is 0.06% higher than that from the single FT loss for the last group. This indicates that the combined FT loss does not improve the performance thus we have used the single FT loss throughout the paper. In terms of paraphrasing the information of the teacher network, the paraphraser which maintains the spatial dimension outperformed autoencoders based methods which use CAE or RAE.

As a second experiment, we compared FT with transferring FitNets-style hints which use full activation maps as in [30]. Table 2 shows the results which verifiy that using the paraphrased information is more beneficial than directly using the full activation maps (full feature maps). In the table, FT gives better accuracy improvement than full-activation transfer (F-ActT). Note that we trained a teacher network from scratch for factor transfer (the last column) with the same experimental environment of [30] because there is no pretrained model of the teacher networks.

## 4.2   CIFAR-100

For further analysis, we wanted to apply our algorithm to more difficult tasks to prove generality of the proposed FT by adopting CIFAR-100 dataset. CIFAR-100 dataset contains the same number of images as CIFAR-10 dataset, 50K (train) and 10K (test), but has 100 classes, containing only 500 images per classes. Since the training dataset is more complicated, we thought the number of blocks (depth) in the network has much more impact on the classification performance because deeper and stronger networks will better learn the boundaries between classes. Thus, the experiments on CIFAR-100 were designed to observe the changes depending on the depths of networks. The teacher network was fixed as ResNet-110, and the two networks ResNet-20 and ResNet-56, that have the

| Student | Teacher | Student | AT | KD | FT | AT+KD | FT+KD | Teacher |
|---|---|---|---|---|---|---|---|---|
| ResNet-56 (0.85M) | ResNet-110 (1.73M) | 28.04 | 27.28 | 27.96 | **25.62** | 28.01 | 26.93 | 26.91 |
| ResNet-20 (0.27M) | ResNet-110 (1.73M) | 31.24 | 31.04 | 33.14 | **29.08** | 34.78 | 32.19 | 26.91 |

| Student | Teacher | $k=0.5$ | $k=0.75$ | $k=1$ | $k=2$ | $k=4$ | CAE | RAE |
|---|---|---|---|---|---|---|---|---|
| ResNet-56 (0.85M) | ResNet-110 (1.73M) | **25.62** | 25.78 | 25.85 | 25.63 | 25.87 | 26.41 | 26.29 |
| ResNet-20 (0.27M) | ResNet-110 (1.73M) | 29.20 | 29.25 | 29.28 | 29.19 | **29.08** | 29.84 | 30.11 |

Table 3: Mean classification error (%) on CIFAR-100 dataset (5 runs). All the numbers are from our implementation.

| Paraphraser | Translator | CIFAR-10 | CIFAR-100 | Number of layers in Paraphraser | CIFAR-10 | CIFAR-100 |
|---|---|---|---|---|---|---|
| Yes | No | 6.18 | 27.61 | 1 Layer [0.07M] | 6.09 | 27.07 |
| No | Yes | 6.12 | 27.39 | 2 Layers [0.22M] | 5.99 | 27.03 |
| Yes | Yes | 5.71 | 26.91 | 3 Layers [0.26M] | 5.71 | 26.91 |
| Student (WRN-40-1[0.6M]) | | 7.02 | 28.81 | Teacher (WRN-40-2[2.2M]) | 4.96 | 24.10 |

Table 4: Left: Ablation study with and without the paraphraser ($k = 0.5$) and the Translator. (Mean classification error (%) of 5 runs). Right: Effect of number of layers in the paraphraser.

same width (number of channels) but different depth (number of blocks) with the teacher, were used as student networks. As can be seen in Table 3, we got an impressive result that the student network ResNet-56 trained with FT even outperforms the teacher network. The student ResNet-20 did not work that well but it also outperformed other knowledge transfer methods.

Additionally, in line with the experimental result in [30], we also got consistent result that KD suffers from the gap of depths between the teacher and the student, and the accuracy is even worse compared to the student network in the case of training ResNet-20. For this dataset, the hybrid methods (AT+KD and FT+KD) was worse than the standalone AT or FT. This also indicates that KD is not suitable for a situation where the depth difference between the teacher and the student networks is large.

## 4.3 Ablation Study

In the introduction, we have described that the teacher network provides more paraphrased information to the student network via factors, and described a need for a translator to act as a buffer to better understand factors in the student network. To further analyze the role of factor, we performed an ablation experiment on the presence or absence of a paraphraser and a translator. The result is shown in Table 4. The student network and the teacher network are selected with different number of output channels. One can adjust the number of student and teacher factors by adjusting the paraphrase rate $k$ of the paraphraser. As described above, since the role of the paraphraser (making $F_T$ with unsupervised training loss) and the translator (trained jointly with student network to ease the learning of Factor Transfer) are not the same, we can confirm that the synergy of two modules maximizes the performance of the student network. Also, we report the performance of different number of layers in the paraphraser. As the number of layers increases, the performance also increases.

## 4.4 ImageNet

The ImageNet dataset is a image classification dataset which consists of 1.2M training images and 50K validation images with 1,000 classes. We conducted large scale experiments on the ImageNet LSVRC 2015 in order to show our potential availability to transfer even more complex and detailed informations. We chose ResNet-18 as a student network and ResNet-34 as a teacher network same as in [30] and validated the performance based on top-1 and top-5 error rates as shown in Table 5.

As can be seen in Table 5, FT consistently outperforms the other methods. The KD, again, suffers from the depth difference problem, as already confirmed in the result of other experiments. It shows just adding the FT loss helps to lower about 1.34% of student network's (ResNet-18) Top-1 error on ImageNet.

## 4.5 Object Detection

In this experiment, we wanted to verify the generality of FT, and decided to apply it on detection task, other than classifications. We used Faster-RCNN pipeline [21] with PASCAL VOC 2007 dataset [5]

| Method | Network | Top-1 | Top-5 |
|--------|---------|-------|-------|
| Student | Resnet-18 | 29.91 | 10.68 |
| KD | Resnet-18 | 33.83 | 12.55 |
| AT | Resnet-18 | 29.36 | 10.23 |
| FT ($k = 0.5$) | Resnet-18 | **28.57** | **9.71** |
| Teacher | Resnet-34 | 26.73 | 8.57 |

Table 5: Top-1 and Top-5 classification error (%) on ImageNet dataset. All the numbers are from our implementation.

| Method | mAP |
|--------|-----|
| Student(VGG-16) | 69.5 |
| FT(VGG-16, $k = 0.5$) | 70.3 |
| Teacher(ResNet-101) | 75.0 |

Table 6: Mean average precision on PASCAL VOC 2007 test dataset.

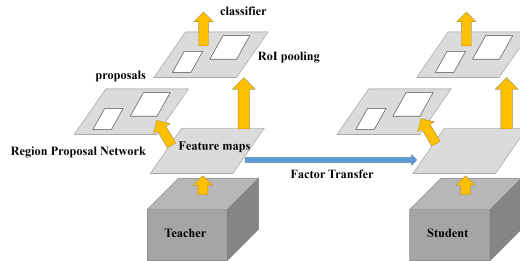

Figure 3: Factor transfer applied to Faster-RCNN framework

for object detection. We used PASCAL VOC 2007 trainval as training data and PASCAL VOC 2007 test as testing data. Instead of using our own ImageNet FT pretrained model as a backbone network for detection, we tried to apply our method for transferring knowledges about object detection. Here, we set a hypothesis that since the factors are extracted in an unsupervised manner, the factors not only can connote the core knowledge of classification, but also can convey other types of representations.

In the Faster-RCNN, the shared convolution layers contain knowledges of both classification and localization, so we applied factor transfer to the last layer of shared convolution layer. Figure 3 shows where we applied FT in the Faster-RCNN framework. We set VGG-16 as a student network and ResNet-101 as a teacher network. Both networks are fine-tuned at PASCAL VOC 2007 dataset with ImageNet pretrained model. For FT, we used ImageNet pretrained VGG-16 model and fixed the layers before conv3 layer during training phase. Then, by the factor transfer, the gradient caused by the $\mathcal{L}_{FT}$ loss back-propagates to the student network passing by the student translator.

As can be seen in Table 6, we could get performance enhancement of 0.8 in mAP (mean average precision) score by training Faster-RCNN with VGG-16. As mentioned earlier, we have strong belief that the latter layer we apply the factor transfer, the higher the performance enhances. However, by the limit of VGG-type backbone network we have used, we tried but could not apply FT else that the backbone network. Experiment on the capable case where the FT can be applied to the latter layers like region proposal network (RPN) or other types of detection network will be our future work.

## 4.6 Discussion

In this section, we compare FitNet [22] and FT. FitNet transfers information of an intermediate layer while FT uses the last layer, and the purpose of the regressor in FitNet is somewhat different from our translator. More specifically, Romero *et al.* [22] argued that giving hints from deeper layer over-regularizes the student network. On the contrary, we chose the deeper layer to provide more specific information as mentioned in the paper. Also, FitNet does not use the paraphraser as well. Note that FitNet is actually a 2-stage algorithm in that they initialize the student weights with hints and then train the student network using Knowledge Distillation.

## 5 Conclusion

In this work, we propose the factor transfer which is a novel method for knowledge transfer. Unlike previous methods, we introduce factors which contain paraphrased information of the teacher network, extracted from the paraphraser. There are mainly two reasons that the student can understand information from the teacher network more easily by the factor transfer than other methods. One

reason is that the factors can relieve the inherent differences between the teacher and student network. The other reason is that the translator of the student can help the student network to understand teacher factors by mimicking the teacher factors. A downside of the proposed method is that the factor transfer requires the training of a paraphraser to extract factors and needs more parameters of the paraphraser and the translator. However, the convergence of the training for the paraphraser is very fast and additional parameters are not needed after training the student network. In our experiments, we showed the effectiveness of the factor transfer on various image classification datasets. Also, we verified that factor transfer can be applied to other domains than classification. We think that our method will help further researches in knowledge transfer.

### Acknowledgments

This work was supported by Next-Generation Information Computing Development Program through the NRF of Korea (2017M3C4A7077582) and ICT R&D program of MSIP/IITP, Korean Government (2017-0-00306).

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
