[Supplementary Material]

# Supplementary Material for Paraphrasing Complex Network: Network Compression via Factor Transfer

**Jangho Kim**
Seoul National University
Seoul, Korea
kjh91@snu.ac.kr

**SeongUk Park**
Seoul National University
Seoul, Korea
swpark0703@snu.ac.kr

**Nojun Kwak**
Seoul National University
Seoul, Korea
nojunk@snu.ac.kr

## A   Implementation details

**Paraphraser:** In all our experiments, we used a simple paraphraser, each of which has three 2-d convolution layers and three 2-d transposed convolution layers, all six layers using $3 \times 3$ kernels, stride of 1, padding of 1, and batch normalization with leaky-ReLU with rate of 0.1 followed by each of the six layers. This means that we do not reduce spatial dimensions (height and width). Instead,  at the second convolution, we only decrease or increase the number of output feature maps according to a paraphrase rate ($k$). Similarly, the second transposed convolution layer is resized to match the output of last group of teacher network ($m$).

In the all experiments except CIFAR-10, we found that the paraphraser without batch normalization is more beneficial than the paraphraser with batch normalization so we did not use batch normalization except CIFAR-10 dataset. However, when training a paraphraser without a batch normalization layer as $l_2$ loss instead of $l_1$ loss, it is necessary to tune the learning rate so that an exploding gradient does not occur.  Besides, to modulate the dimension of student factors $F_S$ to match the dimension of teacher factors $F_T$, the student translator has the same three convolution layers as the paraphraser.

**CIFAR-10 and CIFAR-100**: We trained the paraphraser for the maximum of 30 epochs starting with learning rate of 0.1 because validation loss converges within a few epochs whereas the training loss of the paraphraser takes long to converge as shown in Figure 1, which implies that the network might be overly fitting to the training set. Actually, training a paraphraser for too many epochs slightly diminishes the performance of the student network.

After training the paraphraser with feature maps of the last group, the paraphraser can extract the task specific features (teacher factors) shown in Figure 2 because the higher level layer features output more class specific features compared to lower layers.

At the student network training phase, we started with a learning rate of 0.1, and decayed the learning rate with a factor of 0.1 at 32,000 and 48,000 iterations and completed training at 64,000 iterations, the same way as [1]. The weight decay and the momentum were set to $5 \times 10^{-4}$ and 0.9 respectively, and used a SGD (stochastic gradient descent) optimizer with a mini-batch size of 128 on a single Titan Xp.

$$\mathcal{L}_{FT} = \|\frac{F_T}{\|F_T\|_2} - \frac{F_S}{\|F_S\|_2}\|_p. \tag{1}$$

The performance difference between $l_1$ ($p = 1$) and $l_2$ ($p = 2$) for FT loss is minor (See the Table 1), so we consistently used $l_1$ loss for all experiments in the paper.

**ImageNet:** For the same reason of CIFAR dataset, using learning rate of 0.1, we finished paraphraser training at the maximum of the one epoch. In the student network training, we started with a learning rate of 0.1. The learning rate is decayed by a factor of 0.1 at every 30 epochs, as typical setting in the ImageNet training. We stopped the training process at 90 epochs. We used a weight decay of $10^{-4}$

Figure 1: A training curve of the paraphraser ($k = 0.5$) with ResNet-56 on CIFAR-10.

Figure 2: t-SNE [2] Visualization of the factor space ($k = 0.5$) for ten classes of CIFAR-10 dataset. The teacher network and the student network are ResNet-56 and ResNet-20 respectively.

| Method | Network | CIFAR-10 | CIFAR-100 |
|--------|---------|----------|-----------|
| Student | Resnet-20 | 7.78 | 31.24 |
| FT ($l1$) | Resnet-20 | 6.85 | 29.08 |
| FT ($l2$) | Resnet-20 | 6.88 | 29.18 |
| Teacher | Resnet-56 | 6.39 | – |
| Teacher | Resnet-110 | – | 26.91 |

Table 1: Comparison of mean classification error (%) when using $l1$ loss and $l2$ loss (5 runs).

and a momentum of 0.9. Also, we used a SGD with the mini-batch size of 256 on two GTX 1080 ti cards.

## B  Details of Convolutional autoencoders

**Convolutional autoencoder (CAE):** Similar to the paraphraser, we used a CAE of six layers, each of which has three 2-d convolution layers and three 2-d transposed convolution layers, all six layers using $4 \times 4$ kernels, stride of 2, padding of 1, and batch normalization with leaky-ReLU with rate of 0.1 followed by each of the six layers. In 2-d convolution layers, we reduced spatial dimensions using stride and increase the number of channels with 2 times. On the other hands, in 2-d transposed convolution layers, we expanded the spatial dimensions and decrease the number of channels to match the number of teacher's input channels. To match the factor dimension, the student translator has the same three convolution layers as the CAE. We set $\beta$ of FT loss to $10^2$ in all experiments.

**Regularized autoencoder (RAE):** The architecture of the RAE is equal to the paraphraser with a paraphrase rate $k$ of 2. Since $L_1$ norm regularization is known to produce sparse coefficients and can be robust to irrelevant features (outliers), $L_1$ penalty is applied to the teacher factors extracted from paraphraser.

$$\mathcal{L}_{rec} = \|x - P(x)\|^2 + \alpha\|F_T\|_1 \qquad (2)$$

We set $\alpha$ to $10^{-6}$ in all our experiments.

# C    Training Curves on Datasets

(a) The student and the teacher networks are VGG13 and WRN46-4 respectively.

(b) The student and the teacher networks are ResNet 56 and ResNet 110 respectively.

Figure 3: The training curve of four different algorithms, AT, KD, FT, and basic student. Figure (a) is trained using CIFAR-10, and figure (B) is trained using CIFAR-100.

(a) Top-1 validation error

(b) Top-5 validation error

Figure 4: The training curve of ResNet-18 on ImageNet ILSVRC 2015 dataset. The Top-1 error is the probability that the student network will not predict the correct answer, and the Top-5 error is the probability that there will be no correct answer in the highest five classes out of the softmax output.