[Reviews · NeurIPS 2018]

Reviewer 1



This paper addresses the topic of transfering knowledge from a teacher to a student network, which is a slimmer network, in order to preserve as much as the accuracy as possible. The new technique proposes to use "factors" that are extracted/encoded from the features maps in the last group of the teacher network. These factors are trained in an unsupervised method using reconstruction loss. The network that is trained for this encoding is called the paraphraser network (paraphrases the "knowledge" of the teacher network). In addition, on the student side, there is a translator network whose purpose is to translate the feature maps of the student network to mimic the factors of the teacher network. As such, the student network is trained end-to-end and the loss function has two components: 1) a compoennt corresponding to the task for which the network is trained (e.g., image classification) and 2) a component that corresponds to the difference between the paraphrased teacher factors and translated student factors. This way, the student network is trained not only on achieving good performance for the task at hand but also to mimic the knowledge of the teacher network. The paper presents a series of comprehenisve results studying the performance of several teacher-student networks for CIFAR-10, CIFAR-100 and Imagenet. In addition to their own technique, they compare it to attention transfer and knowledge distillation. Overall the technique shows better results than just training the student network from scratch. There is one case where the student network trained with the "factor" technique slightly overperforms the teacher network (and consistently so even on the parameter tuning results that are shown). The paper also shows results for object detection. While these results are mixed, it shows that there is probably merit in the "factor" teaching technique beyond image classification. I think the idea of extracting factors from the teacher network and teaching the student network to mimic these factors and training both for factor mimicking and task accuracy at the same time are nice ideas. The results section is a bit tedious at points, but looks comprehensive. 242-245 - I didn't understand the lines here. Please rephrase. ----------------- UPDATE after rebuttal: It would be good to include in the paper the longer discussion on differences with FitNets provided in the rebuttal.

Reviewer 2



This paper aims at transferring information between a complex network and a simpler one by using the so called paraphrase. The idea is having a bridge to transfer knowledge (activations) even when the student (smaller network) has different number of features / feature dimensions compared to the 'teacher' This paraphaser is trained in an unsupervised manner to map the existing feature maps / information into the desired shape via an autoencoder. Once trained, this information will be included in the loss function when training the student one via a translator (to adapt sizes). This translator is jointly trained with the student and aims at helping the student to "assimilate the information". - I wonder about the variations with respect k. Seems like this parameter is crucial and fixing this to a value does not consistently provide the best results. What would be a good way to find this value in a different application? For instance table 1 and 3 show a large variation depending on this number. - the text claims several times on the relevance of the factors extracted. How can this be verified? - Some tables contain a combination of FT+KD and this combination does not always result in better results. What is the take home message here? Why this numbers are not always included? - How does this compare to FitNets and the idea of helping with intermediate layers? - If the dimensions are the set equal, (line 155) I guess there could be an ablation study showing the need for those translators, right? Would be nice to see those numbers. - In the experimental setup it is mentioned about l2 reg but then the algorithm seems to use l1. The way is written is very confusing. - In summary, experiments are not really convincing Other details: - My understanding is table represents a reimplementation of state of the art methods, right? Would be great if the comparison could be directly on published numbers for fair comparison and understanding the strength of the contribution. - Tables contain redundant information. For instance Table 6 two bottom rows. As the space is limited, this could be omitted. - Text suggest the proposal also works for object detection. However, numbers do not really support this (marginal increment of 0. 3 - The way results are presented seem to be overclaiming. For instance, Table 2, in my opinion, does not show "much better accuracy" but some improvement. ------------------------------- I appreciate author's feedback for addressing my concerns. I have increased my score accordingly.

Reviewer 3



This paper proposes a simple method on knowledge distillation. The teacher summarizes its state into a low dimensional embedded factor trained with convolution auto-endocoder, then the student learns to fit the embedded factors. - The evaluation seems to suggest that the proposed method is effective compared to existing knowledge transfer method. - The approach is similar to FitNet in nature, except that additional auto-encoding factors are being used. It would be great if authors can provide additional insights into why this is necessary. - Specifically, it might be helpful to train several variants of factor architecture, starting from 0 layers to proposed layers to evaluate the significance of the auto-encoding structure. - As mentioned in the paper, the training of auto-encoder incurs additional cost and the authors should give quantitative numbers of this cost. In summary, this is an interesting paper with good empirical results, it can be improved by addressing the comments and shed more lights on why the proposed works. UPDATE ---------- I have read the author's response, clarifying the relation to FitNet and impact of the auto-encoding structure would help enhance the paper a lot, and I would recommend the authors to do so